# Validation of Binocular Vision and Ocular Surface Assessment Tools in Digital Eye Strain Syndrome: The DESIROUS Study

**DOI:** 10.3390/jpm15050168

**Published:** 2025-04-25

**Authors:** Maria João Barata, Pedro Aguiar, Andrzej Grzybowski, Carla Lança, André Moreira-Rosário

**Affiliations:** 1Ophthalmology Department, Unidade Local de Saúde de São José, 1150-199 Lisbon, Portugal; mjsbarata@gmail.com; 2University of Evora, 7004-516 Evora, Portugal; 3Department of Therapy and Rehabilitation Sciences, Escola Superior de Tecnologia da Saúde de Lisboa (ESTeSL), Instituto Politécnico de Lisboa, 1990-096 Lisbon, Portugal; 4Comprehensive Health Research Center (CHRC), Escola Nacional de Saúde Pública, Universidade NOVA de Lisboa, 1600-560 Lisbon, Portugal; 5Department of Health Strategies, National School of Public Health, NOVA University of Lisbon, 1600-560 Lisbon, Portugal; 6Institute for Research in Ophthalmology, Foundation for Ophthalmology Development, 61-701 Poznan, Poland; ae.grzybowski@gmail.com; 7Division of Science, New York University Abu Dhabi, Abu Dhabi P.O. Box 129188, United Arab Emirates; 8NOVA Medical School, Faculdade de Ciências Médicas, NMS, FCM, Universidade NOVA de Lisboa, 1169-056 Lisboa, Portugal; 9Comprehensive Health Research Center (CHRC), NOVA Medical School, Faculdade de Ciências Médicas, NMS, FCM, Universidade NOVA de Lisboa, 1169-056 Lisboa, Portugal

**Keywords:** digital eye strain, binocular vision, convergence, accommodation, ocular surface

## Abstract

**Background**: To understand if binocular vision disorders are associated with Digital Eye Strain Syndrome (DESS), a study protocol is needed to ensure consistency across observational studies. This study aims to test the feasibility of a protocol to assess DESS, screen time, binocular vision, and dry eye. **Methods:** DESIROUS is an observational cross-sectional study among Polytechnic students at the Lisbon School of Health Technology, Portugal. The protocol includes three questionnaires (Computer Vision Syndrome Questionnaire [CVS-Q], Convergence Insufficiency Symptom Survey [CISS], and Dry Eye Questionnaire version 5 [DEQ-5]), an assessment of visual acuity and binocular vision (cover test for near and distance, stereopsis, near point convergence (NPC), near point accommodation (NPA), accommodative facility, vergence), and the ocular surface break-up tear (BUT) test. The questionnaires were validated using Cronbach’s alpha. Interobserver variability for BUT was assessed using Cohen’s Kappa, Intraclass Correlation Coefficient (ICC), and Bland–Altman analysis involving three observers (A, B, and C), compared against an expert as the gold standard. **Results:** A total of 18 students were included in the validation phase (mean age: 21.50 ± 0.62 years; females: 77.8%). The internal consistency of the CVS-Q (α = 0.773) and the CISS (α = 0.756) was considered good, while the DEQ-5 showed a reasonable internal consistency (α = 0.594). Observer A had the highest agreement with the gold standard (Cohen’s Kappa = 0.710 and *p* < 0.001; ICC = 0.924, *p* < 0.001). **Conclusions:** We provide a protocol to assess binocular vision and the ocular surface, with an emphasis on objective measures while integrating other assessment approaches. Further studies are necessary to validate this protocol, potentially incorporating new measures to enhance its validity across different populations.

## 1. Introduction

The widespread use of social media had a significant impact on day-to-day living, amplifying screen time and increasing the risk of Digital Eye Strain Syndrome (DESS) [1]. DESS is of significant public health concern [2,3,4]. With the surge in online education prompted by the COVID-19 pandemic, both educators and students have substantially increased their screen exposure [5,6]. Research spanning from 2020 to 2023 across multiple countries showed that the prevalence of DESS among university students ranges from 50.8% to 94.5% [7,8,9,10,11,12,13,14]. This high prevalence may be attributed to the simultaneous use of multiple digital devices, often two or three at a time, for extended periods. Such usage increases the visual and cognitive load, making students particularly susceptible to DESS.

DESS is associated with a range of symptoms, including headaches, blurred or double vision, ocular dryness, difficulty focusing, neck and back discomfort, light sensitivity, and color distortion [2,15,16]. Although these symptoms are often transient, they can evolve into persistent manifestations, which have significant economic consequences for digital device-dependent professionals [1,3,17]. A comprehensive ocular evaluation is essential before diagnosing DESS, which should include refractive assessment, binocular vision evaluation, and ocular surface evaluation [18]. It is important to note that uncorrected refractive error acts as a confounding factor. Therefore, a refractive assessment should always be performed [19].

A previous systematic review highlighted the need for further research in frequent users, such as adolescents, as well as patients with dry eye or accommodative/binocular vision anomalies [20]. These groups may be at higher risk of experiencing ocular discomfort symptoms that may be diagnosed as DESS [20]. Although workplace ergonomics and dry eye syndrome have been associated with DESS [21,22,23,24], further research is warranted to elucidate the mechanisms underlying the potential link with binocular vision anomalies. Previous studies suggested a potential association between DESS and binocular vision, specifically anomalies of convergence and accommodation, leading to increased accommodative lag, reduced amplitude, and a potential shift in the near point of convergence [20]. Nevertheless, further research is necessary to ascertain the origins of complaints, as headaches, double vision, and blurriness mirror complaints linked with accommodative and convergence dysfunctions. A recent study conducted on a clinical population found that visual symptoms were not influenced by the number of hours spent using digital devices but rather by the presence of visual dysfunction, including refractive, accommodative, and/or binocular issues [25]. While studies like Cacho et al., 2024 emphasize the role of pre-existing visual dysfunctions, others suggest that extended device use may contribute to DESS symptoms through mechanisms such as prolonged near work or reduced blink rates [25].

Previous studies have assessed the presence of DESS with the Computer Vision Syndrome Questionnaire (CVS-Q) without assessing binocular vision [11,12,14,26,27], introducing bias due to the potential overlap of symptoms caused by binocular vision disorders or ocular surface conditions. Thus, scientific evidence on the link between binocular vision and DESS is limited [20]. In addition, previous studies were mostly cross-sectional and lacking well-defined protocols [28,29,30,31,32,33,34]. Furthermore, previous studies have evaluated a plethora of variables, without assessing binocular vison [28,29,34,35,36,37]. Moreover, numerous studies failed to adjust their models for potential confounding variables. Hence, it is imperative for vision professionals to understand the origins of visual symptoms related with prolonged screen time [1,38,39]. Thus, future research studies should aim to incorporate objective evaluations of binocular vision and the ocular surface, alongside with objective measurements of screen time. Consequently, our study is guided by two main questions: (1) Can objective assessment of binocular vision and ocular surface parameters detect functional changes associated with DESS more reliably than subjective reports alone? (2) Do binocular vision anomalies, particularly those affecting accommodation and convergence, precede or contribute to the onset of DESS, especially in individuals with excessive screen exposure? To explore whether DESS represents a distinct condition or a manifestation of underlying binocular vision anomalies, refractive errors, or dry eye disease, we developed a structured study protocol. This protocol—DESIROUS—integrates both objective clinical assessments and subjective symptom evaluations to improve our understanding, diagnosis, and management of DESS. The primary aim of this pilot study is to assess the feasibility of implementing this standardized protocol, with a particular focus on objective measures related to binocular vision and ocular surface evaluation.

## 2. Materials and Methods

### 2.1. Study Design and Population

The DESIROUS study is an observational cross-sectional study to be implemented among Polytechnic students of health technologies from the Lisbon School of Health Technology (ESTeSL) in Portugal. The study was approved by the ethical committees of the University of Évora (reference number 22090) and ESTeSL (reference number 96-2022). The DESIROUS study was registered on ClinicalTrials.gov under the following identifier: NCT05675475.

Cluster sampling was selected due to data protection limitations imposed by the ethics committee of ESTeSL. ESTeSL offers 9 undergraduate programs (bachelor’s degree programs) in the field of healthcare. The programs span four academic years, which are defined as classes. The final year of each program is dedicated to research and internship. Three programs were selected to be included in the study: Clinical Physiology, Medical Imaging and Radiotherapy, and Orthoptics and Vision Sciences. This was due to the perceived higher utilization of digital devices in these programs.

### 2.2. Inclusion and Exclusion Criteria

The DESIROUS study inclusion criteria for sample selection encompassed first-, second-, and third-year students of healthcare technologies programs at ESTeSL, as these years are characterized by a heavier curriculum and continuous study load. Additionally, only students aged between 18 and 35 years will be considered eligible to mitigate the confounding factor of presbyopia. Students with history of ocular surgery, strabismus, nystagmus, or amblyopia will be excluded.

### 2.3. Informed Consent

Prior to their participation, all subjects will be required to sign an informed consent form developed in accordance with the principles set forth in the Helsinki Declaration and the Oviedo Convention. The informed consent document provides comprehensive information about this study, including its circumstances, precise nature, and background. Furthermore, the document elucidated the methodologies employed, the confidentiality, and anonymity of the data.

### 2.4. Sample Size Calculation

The study conducted by Cantó-Sancho et al. on university students provided the basis for estimating the sample size for the DESIROUS study [40], showing that 76.6% of the Spanish population had DESS. No published studies were identified regarding the prevalence of DESS among university students in Portugal. A 95% confidence interval for the population prevalence indicates that 288 individuals need to be recruited to achieve an estimation precision of 5 percentage points, which corresponds to a range of 70% to 80% [41]. A total of 300 subjects will be enrolled in the main study to account for potential withdrawals (Figure 1). Here, we present the validation of the instruments that was conducted on a sample of one class from the Orthoprosthetics degree program.

### 2.5. Study Procedures and Schedule of Assessments

The main study visit comprises two distinct methodological approaches. Firstly, three questionnaires will be administered online in Google Forms. Secondly, participants will undergo an orthoptic assessment and ocular surface assessment using the study instruments outlined below (Figure 1).

All students from the selected class (*n* = 26) were invited to participate in the pilot study (Figure 1). Data collection was carried out between 2 February and 3 March 2023. Overall, 18 of 26 third-year Orthoprosthetics students (69.2%) responded to the first phase of the pilot study—questionnaire validation phase. The non-responders (*n* = 8; 30.8%) were working students or those who did not attend classes regularly. Subsequently, 10 students were randomly selected to participate in the second phase of the pilot—ocular surface evaluation and assessment of binocular vision. The mean age of the included students was 21.5 ± 0.62 years (77.8% were females).

### 2.6. Study Instruments

The present study sought to validate the Portuguese-language questionnaires specifically for university health technology students and to evaluate the intra-operator consistency of tests that rely on operator analysis within the scope of ocular surface assessment, thereby strengthening the robustness of future study protocols.

### 2.7. Questionnaires

Three questionnaires, translated and validated for the Portuguese language and culture, were selected through a comprehensive literature review [42,43,44,45]. The Computer Vision Syndrome Questionnaire (CVS-Q) was designed to assess the presence of visual symptoms associated with prolonged computer use, enabling the diagnosis and quantification of symptoms of DESS. The Convergence Insufficiency Symptom Survey (CISS) was designed to enable the identification of the frequency and severity of visual discomfort symptoms in individuals with convergence insufficiency. The abbreviated version of the Dry Eye Questionnaire, version-5 (DEQ-5), evaluates signs and symptoms of dry eye. As previously mentioned, the symptomatology of DESS can overlap with other binocular dysfunctions and ocular surface abnormalities. Therefore, a comparison between the 3 questionnaires and an objective assessment of binocular vision and ocular surface will be made to understand the origin of the symptoms.

### 2.8. Computer Vision Syndrome Questionnaire

The CVS-Q is an instrument that assess and quantifies symptoms associated with Computer Vision Syndrome (CVS) [4]. The instrument employs a Rasch rating scale model to analyze 16 symptoms. The frequency of each symptom is rated on a scale of 0 to 3, with 0 representing never and 3 representing nearly daily. The intensity of each symptom is rated on a scale of 0 to 3, with 0 representing moderate and 3 representing very intense. The frequency and intensity data are recoded to calculate the severity of each symptom, resulting in a total score. A total score of 6 or higher indicates that the individual exhibits symptoms of CVS [4] (Table 1). This questionnaire was translated and validated for the Portuguese language and culture in 2020, in a population of workers using computers for more than six hours per day [42].

### 2.9. Convergence Insufficiency Symptom Survey

The CISS was developed by the Convergence Insufficiency Treatment Trial (CITT) group to evaluate the impact of the treatment on symptoms associated with convergence insufficiency (CI). In a previous study, this questionnaire was validated and considered a reliable tool suitable for clinical use or as an outcome measure for research studies involving adults with CI [46]. The CISS evaluates the presence and severity of symptoms related to CI. It comprises fifteen Likert-scale questions, with response options ranging from “never“ to “always”, scored from 0 to 4 for various typical symptoms of this binocular vision issue. The total of all responses provides a final score of symptomatology for each individual ranging from 0 (no symptoms) to 60 (considered severely symptomatic). In adults, absence of symptoms of CI is classified by scores below 21, while CI is identified with a score of 21 or higher [46] (Table 1). This questionnaire was selected for its extensive clinical utilization, facilitating the diagnosis and monitoring of treatment outcomes in CI [47,48,49,50]. Additionally, this questionnaire was translated and validated for the Portuguese language and culture in 2013, for university students in the health sciences, as well as science and engineering fields, with a mean age of 21.79 ± 2.42 years [45].

### 2.10. Dry Eye Questionnaire

The DEQ-5 is a condensed version of the original DEQ comprising five questions designed to assess ocular discomfort, dryness, and tearing, which are all indicative of dry eye. It is one of the two instruments recommended by the Tear Film and Ocular Surface Society Dry Eye Workshop II Diagnostic Methodology Report [50]. Two questions assess the intensity of the symptoms experienced, with 0 indicating that the symptom has never been experienced and 1 to 5 indicating an increasing level of severity, from mild to very intense. Three questions evaluate the frequency of signs and symptoms, with responses ranging from 0 to 4, corresponding to “Never”, “Rarely”, “Sometimes”, “Often”, and “Constantly”. The aggregation of respondents’ answers results in a final score, which ranges from 0 to 22 points. The questionnaire outcomes allow the differentiation of patients suspected of having dry eye disease without Sjögren’s syndrome (greater than 6 points) and those suspected of having dry eye due to Sjögren’s syndrome (greater than 12 points) from those without alterations (less than or equal to 6 points) [51] (Table 1).

The full version of the DEQ comprises 21 items with 58 questions and assesses symptoms by determining frequency, daytime severity, and intensity on a typical day during a one-week recall period. This questionnaire also includes inquiries about the time of day symptoms worsened, impact on daily life activities, medications, allergies, dry mouth, nose or vagina, treatments, and overall patient assessment [52]. This comprehensive questionnaire has been used in epidemiological and clinical studies, and the full version of the DEQ was initially chosen due to its broader scope [42,43,44,45]. However, the length of the questionnaire would make it impractical for students to participate and would unnecessarily prolong this study’s duration. The DEQ-5 was found to be more applicable in clinical settings due to its smaller size, cross-cultural translation, and validation. The questions are related to symptoms that occurred during previous month, assessing both the frequency and the intensity of symptoms [7]. A recent study conducted in an African population applied Rasch analysis to the DEQ-5 and found satisfactory psychometric properties for clinical use [53].

### 2.11. Binocular Vision and Screen Time Assessment

An assessment of binocular vision and screen time data were included in the protocol consisting of the following:Subjective data will be collected by asking the amount of time (in hours) spent using various digital devices, including desktops, laptops, tablets, and smartphones. Additionally, data regarding the number of hours of sleep per day, averaged over the past week (7 days), will also be gathered.Objective data on screen time will be collected by examining the participant’s mobile phone usage (average daily time over the past week, last 7 days), phone model, and screen size. Smartphones (Android and iOS systems) have this information available in the digital well-being area for Android devices and in screen time settings for iOS devices. In addition to this information, participants will be asked about the electronic device (desktop, laptop, tablet, smartphone) used most frequently per day and if they own a smartwatch.Evaluation of objective refraction without cycloplegia, using the automatic refractometer GR-21 GRAND SEIKO (Japan).Distance visual acuity assessment with refractive correction using the CSV-1000 ETDRS provides a full range of LogMAR testing (1.0 to −0.3) at a test distance of 8 feet using standardized luminance, 85 cd/m^2^.Identification of oculomotor deviations using the cover test for near and distance vision, with the Lang fixation cube (LANG-STEREOTEST AG) at 40 cm for near and a distant fixation point for far. An opaque cover spoon will be employed at both distances.Evaluation of near stereopsis using the Random Dot Butterfly stereotest (Stereo Optical Co., Chicago, IL, USA) graded circle test at 40 cm, 800 to 40 s of arc.Assessment of near point of convergence (NPC) and accommodation (NPA) using the RAF (Royal Air Force) ruler (Haag-Streit, UK). For measuring the NPC, the target used is the fixation point on the near point card. The card will be brought from a distance of 50 cm along the facial midline in free space, moving approximately 2 cm/s towards the participant’s nasal bridge. The card will be stopped when the participant reports seeing double or when the examiner obvers any eye deviation. This measurement will be repeated three times. The measurement of the NPA is similar to that of the NPC. However, in this test, the target used is the N5 horizontal line, and the card is stopped when the participant reports that the letters are blurred.Assessment of accommodative facility using Flippers ±2.00 Diopter and near visual acuity chart “1” in LogMAR sizes for testing at 16 inches (40 cm). The patient focuses on the 0.1 visual acuity line on the near vision scale, and the examiner counts how many cycles the participant can complete in one minute. Each cycle consists of one positive lens and one negative lens. The examiner places the flipper, and when the vision becomes clear, the participant indicates this quickly by saying “now”, after which the examiner alternates the lens.Assessment of fusional amplitudes in space, convergence, and divergence for near and distance vision using horizontal prism bar and for near Lang fixation cube (LANG-STEREOTEST AG) and for far distant fixation point.

The normative values for each variable of the binocular vision assessment are detailed in Table 2.

The data collection for the main study will be conducted by three operators. All the operators had training on how to perform the binocular vision to ensure consistency in all the measurements. Potential environmental confounding factors, including lighting and posture, were controlled throughout this study. All data collection took place in the same examination room under consistent lighting conditions. Both participants and examiners adhered to standardized postures throughout the entire procedure.

Binocular vision assessment will be classified as either normal or abnormal. A student that falls within the values shown in Table 2 will be classified as normal, while students with at least 1 value outside the normal range will be classified as abnormal.

### 2.12. Ocular Surface Assessment

Tear film break-up time (BUT) test is a technique that uses a slit-lamp device and sodium fluorescein strips to measure the stability of the tear film. The diffuse technique with the cobalt blue filter is used during the measurements. In general, more than 10 s is considered as normal and less than 10 s as dry eye [61]. A weak tear film is indicated by a fast tear break-up time; the longer it takes, the more stable the tear film.

Given the presence of three operators in data collection, it was crucial to ensure the absence of inter-operator variability. To establish interobserver validation, an expert in ocular surface was consulted as the gold standard.

Initially, training on performing BUT measures was conducted, followed by the pre-validation stage. The BUT test was recorded on video using a camera attached to a slit lamp. Subsequently, these videos were anonymized and sent to the three members of the data collection team (A, B, and C) and an expert in ocular surface with 20 years of experience, the gold standard for assessment. The results of the videos assessments were compared and statistically analyzed to evaluate their level of agreement.

### 2.13. Statistical Analysis

Questionnaire validation was assessed with the Cronbach’s alpha coefficient, which measures the degree of variability and reliability of internal consistency within a scale. The Cronbach’s alpha [62] represents the average of correlations among items within an instrument, reflecting the correlation between questionnaire responses, and values range from 0 to 1 (values closer to 1 suggest higher internal consistency; values below 0.70 suggest low internal consistency; 0.70 is a minimum acceptable value) [63]. A maximum value exceeding 0.90 may indicate redundancy or duplication, with several similar questions assessing the same object. Thus, alpha values between 0.80 and 0.90 are considered ideal [63].

For the BUT measures, the Cohen’s Kappa coefficient was used to measure the level of agreement between the expert and each of the data collection members. In addition, Fleiss’ Kappa was also employed to gain an overall understanding of the agreement among the 3 participants. Fleiss’ Kappa is derived from Cohen’s Kappa and is considered the most suitable coefficient for evaluating the degree of agreement among three or more examiners, particularly when the evaluated scale has many categories. Kappa values can range from −1 to +1. A value of −1 indicates total disagreement, 0 indicates agreement equivalent to chance, and values above 0 represent increasing agreement among the evaluators, up to the maximum value of +1, indicating perfect agreement. Subsequently, the analysis focuses on each observer compared to the expert, for which three stages of statistical analysis are performed, Intraclass Correlation Coefficient (ICC) and Bland–Altman analysis. The ICC value ranges from 0 to 1, with values below 0.5 indicating low reliability, values between 0.5 and 0.75 indicating moderate reliability, values between 0.75 and 0.9 indicating good reliability, and values above 0.9 indicating excellent reliability.

The Bland–Altman analysis was used to assess the agreement between the 3 observers and the expert. The limits of agreement were calculated using the mean and standard deviation of the differences between the observers and the expert, allowing the construction of the Bland–Altman scatter plot. In this plot, a high degree of agreement is shown by the reduction in dispersion of points and their proximity to the line representing the mean bias. Conversely, a low degree of agreement is indicated by the high dispersion of points and their distance from the line representing the mean trend. Data analysis was conducted using IBM^®^ SPSS Statistics^®^, version 27, with a significance level of 0.05.

## 3. Results

The most frequently used digital device was the smartphone (5.33 ± 2.40 h per day), followed by the laptop (2.06 ± 1.78 h per day or week). Additionally, the students’ subjective screen time was compared to the objective screen time, revealing only a slight overestimation (5.33 ± 2.40 versus 4.89 ± 1.91 h).

The results of the questionnaires showed that 77.8% (*n* = 14) of participants exhibited symptoms of DESS, and 27.8% (*n* = 5) showed responses indicative of convergence insufficiency. In contrast, the analysis of the DEQ-5 revealed that only 38.9% (*n* = 7) of participants exhibited symptoms of dry eye disease (DED).

### 3.1. Validation of the Questionnaires

Initially, the Cronbach’s alpha for the questionnaires’ answers was calculated based on standardized items, yielding a value of 0.869. Subsequently, individual analysis of each questionnaire was conducted to assess the overall internal consistency (Figure 2) and the consistency of each question item.

Due to the lower consistency of DEQ-5, an evaluation of the “Item-Total Statistics” was conducted, which identified that removing question number 5 (“In the last month, on a normal day, how often did you feel or have your eyes watery?”) would increase the alpha value to 0.710, a value on the threshold of acceptable internal consistency.

Additionally, participants were queried regarding the duration and difficulty of the three questionnaires. All respondents indicated that the duration and difficulty were appropriate. No missing values were identified during the analysis of the questionnaire data.

The digital format responses provided greater validity control and was more secure than the paper version. In the paper-based questionnaire, there were two instances of missing data due to non-responses regarding the intensity of reported symptoms in the frequency section in the CVS, showing only 80% of valid answers. This issue did not occur in the online version, which required mandatory responses (100% of the answers were valid).

### 3.2. Binocular Vision and Ocular Surface Assessment

From the ten students randomly selected for the pilot study, only nine consented to be assessed. Six students exhibited symptoms of DESS, three had symptoms of CI, and three exhibited dry eye disease (DED) symptoms. Average visual acuity (0.13 ± 0.14 LogMAR), near stereopsis (44.44 ± 7.27 *s* of arc), and NPC (7.56 ± 3.25 cm) were only slightly reduced, while NPA (11.67 ± 2.83 D), binocular accommodative facility (6.11 ± 4.60 cpm), and fusional amplitudes (Positive Fusional Vergence (PFV) for far = 21.00 ± 10.42 DP; PFV for near = 31.33 ± 10.94 DP; Negative Fusional Vergence (NFV) for near = 12.56 ± 6.23 DP; NFV for far = 8.50 ± 2.07 DP) were within normal ranges (Table 3). In the objective assessment of binocular vision, seven participants were identified as having binocular vision anomalies, with five of these cases presenting convergence insufficiency. Among those with binocular vision anomalies, five tested positive on the CVS-Q, whereas only two were positive on the CISS.

In the BUT measurements, only Observer A and the expert exhibited substantial agreement, with Cohen’s Kappa ranging from 0.61 to 0.80 (Cohen’s Kappa = 0.710 and *p*-value < 0.001). For Observer B (Cohen’s Kappa = −0.127; *p*-value = 0.46) and Observer C (Cohen’s Kappa = −0.094; *p*-value = 0.59), the agreement with the expert was negative and not statistically significant. The interobserver reliability assessed by the ICC revealed that Observer A exhibited an ICC of 0.924 (*p* < 0.001). Conversely, the ICC between Observer B and Observer C with the expert showed a low level of agreement, with ICC values of 0.466 and 0.380 respectively (both *p* < 0.001).

Bland–Altman analysis is presented in Figure 3A–C. For Observer A (Figure 3A), all values were within the levels of agreement, except for one. The dispersion between the points and the line representing the mean bias was not considerable (mean ± standard deviation [SD] = −0.059 ± 0.429; limits ranging from −0.899 to 0.782). For Observer B (Figure 3B), all values were within the levels of agreement, and the dispersion between the points and the line representing the mean bias was not significant (mean ± SD =0.118 ± 0.928; limits ranging from −1.700 to 1.936). Similar results were observed for Observer C (Figure 3C), with all values within the levels of agreement, and the dispersion between the points and the line representing the mean bias was minimal (mean ± SD = 0.235 ± 0.903; limits ranged from −1.535 to 2.006).

## 4. Discussion

The literature review indicated the need to develop a protocol for the assessment of binocular vision and the ocular surface to ascertain their role in DESS. We have conducted a feasibility study to test the implementation of such protocol, which may facilitate comparison across different observational studies and ensure applicability in both clinical and scientific research contexts. We found notable inconsistencies in the subjective assessment of visual symptoms across studies. While most studies used only a single questionnaire [28,29,31,34,37,39], two studies employed two questionnaires, with both including CVS-Q [35,64]. The CVS-Q emerged as the most frequently used tool; however, the overall variability in the selection and application of questionnaires limits comparability between studies. One of the key strengths of the present study is the inclusion of three questionaries and an objective assessment of binocular vision and the use of objective metrics for smartphone screen time, collected directly from each participant’s device. The DESIROUS protocol integrates a comprehensive set of instruments, including questionnaires, binocular vision assessment methodologies, and ocular surface evaluation, all supported by validated data collection tools.

The present study revealed that two of the three included questionnaires, CVS-Q and CISS, exhibited good internal consistency, whereas the DEQ-5 showed only moderate internal consistency. The original Spanish CVS-Q questionnaire demonstrated sensitivity and specificity above 70% and a Cronbach’s alpha of 0.78 [4]. A 2024 literature review on DESS and binocular vision changes revealed that the CVS-Q was the most frequently used tool [28,31,33,40] due the straightforward assessment of the syndrome prevalence. Accordingly, this questionnaire was selected for inclusion in the DESIROUS study. Another contributing factor was its adaptation and translation into Portuguese [42,65], which demonstrated good internal consistency with Cronbach’s alpha values of 0.87 for frequency assessment and 0.87 for intensity assessment [42].

Rouse et al. (2004) demonstrated that the CISS exhibited excellent internal consistency, with a Cronbach’s alpha of 0.96. Furthermore, the questionnaire exhibited high sensitivity (97.8%) and specificity (87%) in detecting CI in a population of young adults aged 19 to 30 [46]. Upon translation and cultural adaption to Portuguese in 2014, the CISS exhibited good internal consistency with a Cronbach’s alpha of 0.89. Additionally, it demonstrated high reproducibility as a tool for measuring visual discomfort associated with near-vision tasks, similar to other studies [45]. A recent study applying Rasch analysis to the structure of the CISS identified that certain categories within the questionnaire require refinement to enhance its performance. Nonetheless, the authors concluded that the CISS remains a valid and reliable tool for measuring the symptoms it assesses [66]. We performed a Rasch analysis of CISS. However, the analysis failed to produce a stable solution regarding the quality of the items within the scale and the resulting measurement model. Initially, we observed that the Rasch analysis software excluded 87% (13 out of 15) of the items. This outcome is likely due to the small sample size and the limited variability in item responses within the scale. The small sample size resulted from selecting a class of 26 students, of whom only 18 agreed to participate in this study. Given these limitations, we opted to use Cronbach’s alpha alone, as it remains a valid approach for assessing internal consistency in our context. However, Rasch analysis could be advantageous in the future, and we plan to conduct it in the main study, where a larger and more diverse sample will allow for a more robust psychometric analysis.

The CISS, specifically designed for CI, does not allow for the identification of other binocular vision anomalies, as the symptoms of accommodative and non-strabismic binocular dysfunctions are varied and often overlap, with no clear consensus on which should be considered for the diagnosis of each condition [67]. Recently, the Symptom Questionnaire for Visual Dysfunctions (SQVD) has been developed to identify symptoms associated with various binocular vision disorders [68]. However, it has not yet been translated into Portuguese. In the future, it would be valuable to translate and culturally adapt this questionnaire for Portuguese and university students, as it was validated in a clinical population.

In the Portuguese translation of the DEQ, internal consistency was not statistically evaluated with Cronbach’s alpha [43]. Instead, content validity was assessed by calculating the percentage agreement for each item across three translations, resulting in a high agreement rate of 81.5% among members from the first and second evaluation committees. Gross et al. revealed that the interobserver reliability of the translated DEQ-5 varied from 0.584 to 0.813, where question 2b (“When your eyes were dry, how intense was this dryness at the end of the day, in the two hours following bedtime?”) exhibited the lowest reliability [44]. In terms of internal consistency, the questionnaire had a Cronbach’s alpha of 0.89, indicating a high level of internal consistency [44]. The lower alpha value observed in this pilot study may be attributed to the small sample size. All items were retained to be tested in the main study to maintain comparability with the existing literature. However, future studies with larger and more diverse populations will be necessary to reassess internal consistency. These studies may also include item-level analyses to identify and refine potentially problematic items, thereby improving the questionnaire’s reliability in the context of DESS.

A comparison of the internal consistency values obtained in the present study with the original questionnaires and their translations into Portuguese revealed that, although the internal consistency was good, values were slightly lower than those obtained originally. This divergence may be attributed to the characteristics of the sample population, as the previous study included older individuals than that of the present study who were less susceptible to contact with electronic devices, which may have rendered certain items potentially less suitable for their age group. These findings may also be attributed to the limited sample size.

It is important to note that questionnaires have disadvantages, particularly memory bias and the subjectivity of responses. These instruments assess the frequency and intensity of symptoms, yet not all individuals have the same ability to accurately recall past events. Moreover, the measurement scale is influenced by the inherent subjectivity of each respondent’s perception. For this reason, although questionnaires provide valuable indicators for assessing certain conditions, it is essential to complement the information obtained with objective measurements, ensuring a more precise and robust evaluation.

To evaluate binocular vision, previous studies have focused on accommodation, although the tests used varied. Most studies evaluated BAF, NPA [28,29,31,35,37,64], and accommodative posture. A limited number of studies have assessed fusional vergence [30,33] and the NPC [28,30,37]. A strength of our protocol is its comprehensive approach, as it includes all these assessments—accommodation, fusional vergence, and NPC—ensuring a more complete evaluation of binocular vision. While the cover test was performed to identify the heterophoria, a limitation of the current study is the lack of prismatic cover test measurements, which would enable the quantification of the heterophoria. Future studies should test the accuracy of adding the prismatic cover test measurements to the protocol.

The analysis of interobserver variability for the BUT test revealed that Observer A had the highest and statistically significant results, indicating excellent agreement with the expert. Caution should be exercised when using measures from non-expert operators given the considerable discrepancies observed between operators for the BUT test. A low ICC value may be indicative of not only a low degree of agreement among assessors but also a lack of variability among individuals in the sample, a small sample size, or a small number of observers [69]. The Bland–Altman analysis revealed that all observers demonstrated a good level of agreement, as the dispersion of points was reduced and relatively close to the line representing the mean bias. However, Observer A exhibited the most reliable agreement with the gold standard, as evidenced by the narrower limits, suggesting more accurate and reliable measurements. Based on these findings, in the main study, video recordings of the BUT test will be taken with a camera and will be evaluated by Observer A to ensure the reliability of all measurements. Given that we found variability in the administration of the BUT test, future studies should incorporate rigorous examiner training protocols and consider the use of video-assisted documentation to ensure greater consistency and reliability in objective testing.

Given that assessments of symptoms rely on subjective questionnaires, this study highlights the need to develop further studies with more objective measures to improve research accuracy and reliability of study protocols. In the future, this protocol could be optimized by removing or incorporating other instruments to improve its validity and accuracy. In particular, efforts will be made to improve standardization in the main study, including (1) structured examiner training sessions using standardized video demonstrations and competency-based checklists; (2) calibration procedures across examiners to ensure consistent adherence to the examination protocols; and (3) an evaluation of the feasibility of video-assisted documentation to allow for post hoc quality control and improved reproducibility.

### Limitations of the Study

This study has several limitations that should be considered for future protocol improvements. A limitation of this study is the relatively small sample size. However, as a pilot, the primary aim was to test the feasibility and robustness of the study protocol. Future studies involving larger and more diverse populations will be essential to enhance statistical power and assess the protocol’s applicability across broader demographic and clinical settings. The questionnaire chosen to assess convergence insufficiency is widely used internationally; however, some studies have raised concerns about its suitability as a screening or diagnostic tool for CI [70,71]. Furthermore, a recent Rasch analysis identified aspects of the questionnaire that require refinement to enhance its performance [66].

In our pilot study, the validation of the questionnaires was limited to an assessment of internal consistency due to the sample size being insufficient for a full psychometric analysis. Future studies with larger and more diverse samples are needed to enable comprehensive psychometric validations, including assessments of construct validity, test–retest reliability, and factor structure. To further enhance the comprehensiveness of symptom profiling in future phases, additional validated tools, such as SQVD, may be incorporated. These instruments could provide a more nuanced understanding of the range and severity of visual symptoms associated with DESS, particularly those related to binocular vision. Although the SQVD is not yet available in Portuguese, its inclusion in the protocol could add significant value, pending appropriate translation and cultural adaptation.

In future phases of the study, it is important to consider the addition of a longitudinal follow-up to monitor DESS symptoms and visual and binocular function changes, particularly in relation to sustained screen exposure. This will help clarify whether certain functional markers, such as accommodation or tear film instability, may serve as early indicators of symptom development.

It is therefore recommended that the protocol be refined in line with ongoing scientific research. Despite these limitations, the results contribute to the validation of the study instruments, providing a more robust foundation for future studies using this methodology and serving as a basis for further research in the field.

## 5. Conclusions

The DESIROUS protocol aims to employ methods to ensure consistent data collection procedures, including a combination of objective and subjective measures. This pilot study supports the feasibility of implementing the DESIROUS protocol to assess binocular vision and ocular surface parameters. These preliminary findings serve as a foundation for a larger study aimed at exploring these associations in greater depth. The overarching goal is to establish clinical guidelines that assist eye care professionals in incorporating objective assessments into clinical practice, thereby enabling more personalized and effective management of visual complaints. This study contributes to the advancement of personalized medicine by integrating both subjective and objective measures to better characterize individual visual profiles associated with DESS. The DESIROUS protocol offers a structured framework for identifying potential functional biomarkers—such as binocular vision anomalies or ocular surface anomalies—that may help distinguish true DESS from other coexisting or underlying conditions requiring clinical intervention, such as convergence insufficiency, uncorrected refractive errors, or dry eye disease. By facilitating early and more accurate diagnosis, this approach may enable tailored treatment strategies, reduce misdiagnosis, and support targeted patient management.

## Figures and Tables

**Figure 1 jpm-15-00168-f001:**
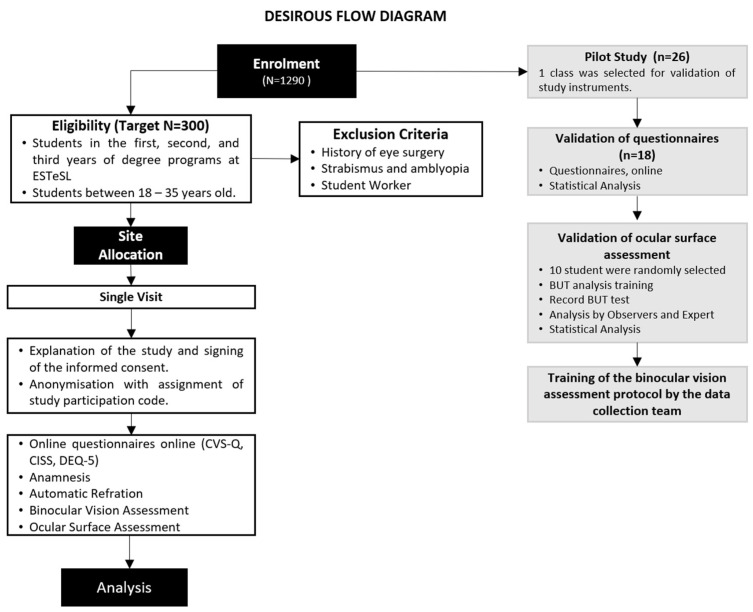
DESIROUS flow diagram. Legends: ESTeSL: Lisbon School of Health Technology; CVS-Q: Computer Vision Syndrome Questionnaire; CISS: Convergence Insufficiency Symptom Survey; DEQ-5: Dry Eye Questionnaire version 5; BUT: tear film break-up time.

**Figure 2 jpm-15-00168-f002:**
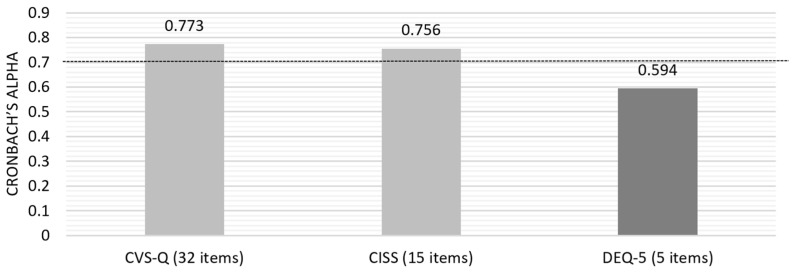
Cronbach’s alpha for the three questionnaires included in the pilot study.

**Figure 3 jpm-15-00168-f003:**
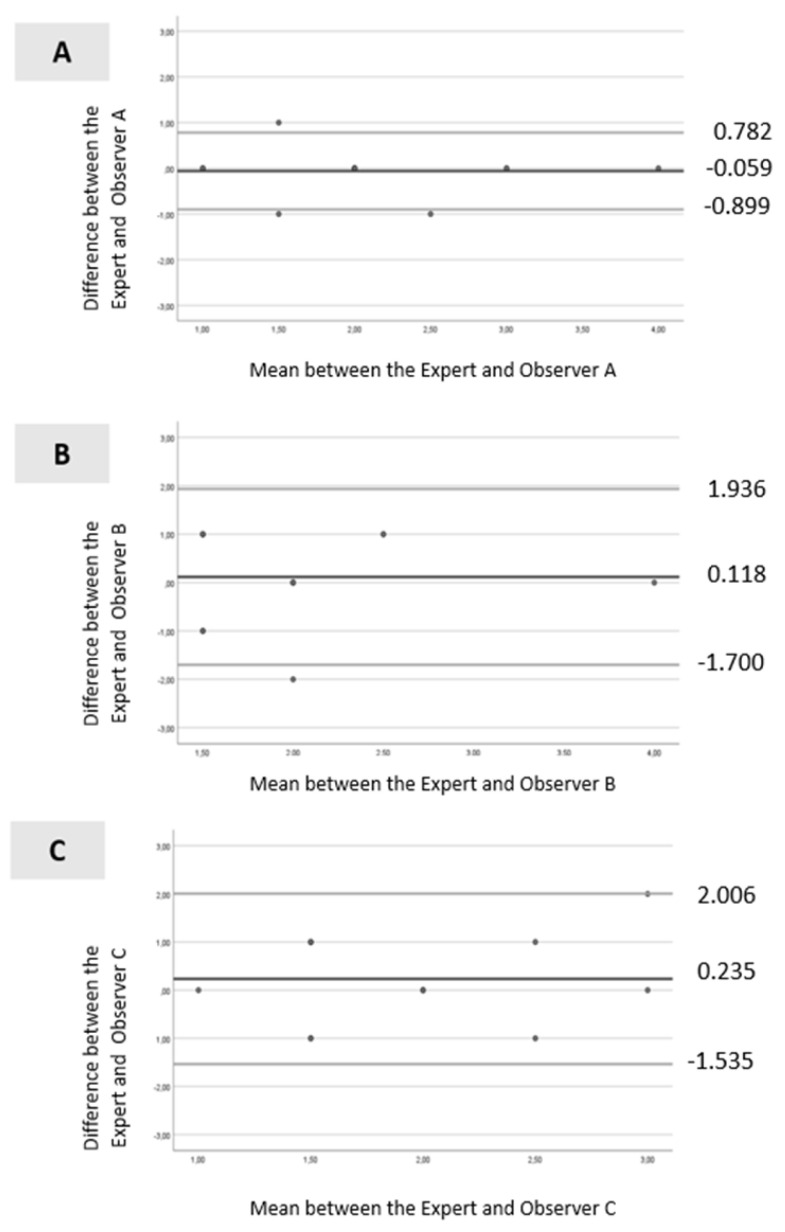
Bland–Altman scatter plot: (**A**)—between the expert and Observer A; (**B**)—between the expert and Observer B and (**C**)—between the expert and Observer C. Legend: black line (--): mean line; gray line upper (--) = mean difference + 1.96 x SD difference; gray line lower (--) = mean difference—1.96 x SD difference.

**Table 1 jpm-15-00168-t001:** Summary of the characteristics and scores of the questionnaires applied, CVS-Q, CISS, and DEQ-5.

	Questionnaires
CVS-Q	CISS	DEQ-5
**Purpose**	Determine the visual health of computer users	Identifying individuals with convergence insufficiency	Determine the presence of dry eye disease
**Items**	16 × 2	15	5
**Frequency**Response Type and Points	“Never”	0	“Never”	0	“Never”	0
“Occasionally”	1	“Infrequently”	1	“Rarely”	1
“Often/Always”	2	“Sometimes”	2	“Sometimes”	2
	“Very often”	3	“Often”	3
“Always”	4	“Constantly”	4
**Intensity**Response Type and Points	“Moderate”	1	Non-applicable	“Never felt”	0
“Intense”	2	“Not intense”	1
	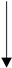	2
3
4
“Very intense”	5
**Score**Value and Interpretation	<6	Asymptomatic CVS	<21	Absence of CI	≤6	Normal
≥6	Symptomatic CVS	≥21	CI	>6	DED
				>12	Suspect SS

Legends: CVS-Q: Computer Vision Syndrome Questionnaire; CISS: Convergence Insufficiency Symptom Survey; DEQ-5: Dry Eye Questionnaire version 5; CVS: Computer Vision Syndrome; CI: convergence insufficiency; DED: dry eye disease; SS: Sjögren’s Syndrome.

**Table 2 jpm-15-00168-t002:** Summary of normative values for the binocular vision in the DESIROUS protocol.

Instrument	Normative Values [54,55,56,57,58,59,60]
Binocular Vision Assessment	
Cover test	ExophoriaEsophoria
Visual acuity (LogMAR)	≤0.0
Near stereopsis	≤40”
Near point of convergence	<6 cm
Near point of accommodation	Hofstetter’s formula = 15 − (0.25 × age)
Binocular accommodative facility	≥3 cpm
PFV for far	20/25 PD
PFV for near	30/40 PD
NFV for far	4/8 PD
NFV for near	12/16 PD

Legends: cm; centimeters; cpm: cycles per minute; PD: Prismatic Diopter; PFV: Positive Fusional Vergence; NFV: Negative Fusional Vergence.

**Table 3 jpm-15-00168-t003:** Results from the objective assessment of refractive error, visual acuity, binocular vision, and screen time.

	Mean ± SD
Age (y)	21.40 ± 0.84
Spherical equivalent (D)	
RE	−1.39 ± 1.38
LE	−1.06 ± 1.27
Sleep hours (h/d)	6.94 ± 0.58
Desktop hours (h/d)	1.56 ± 1.33
Laptop hours (h/d)	2.06 ± 1.78
Tablet hours (h/d)	0.44 ± 1.33
Smartphone hours (h/d)	5.33 ± 2.40
Objective smartphone screen time (h/d)	4.88 ± 1.91
VA (LogMAR)	
RE	0.13 ± 0.14
LE	0.06 ± 0.07
Near stereopsis (second of arc)	44.44 ± 7.27
NPC (cm)	7.56 ± 3.25
NPA (D)	11.67 ± 2.83
Accommodative facility (cpm)	6.11 ± 4.60
Vergences (PD)	
PFV for far	21.00 ± 10.42
PFV for near	31.33 ± 10.94
NFV for near	12.56 ± 6.23
NFV for far	8.50 ± 2.07

Legends: SD: standard deviation; y: years; D: Diopter; RE: Right Eye; LE: Left Eye; h/d: hours per day; VA: visual acuity; NPC: near point convergence; cm: centimeters; NPA: near point accommodation; cpm; cycles per minute; PD: Prismatic Diopter; PFV: Positive Fusional Vergence; NFV: Negative Fusional Vergence.

## Data Availability

The datasets used and/or analyzed during the current study are available from the corresponding author on reasonable request.

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
