# Peer review of "Validation of Binocular Vision and Ocular Surface Assessment Tools in Digital Eye Strain Syndrome: The DESIROUS Study"

_jpm, 2025, doi:10.3390/jpm15050168_

Round 1
Reviewer 1 Report
Comments and Suggestions for Authors
General Comments
This study presents an important validation of assessment tools for Digital Eye Strain Syndrome (DESS), focusing on binocular vision anomalies and ocular surface conditions. The study follows a structured observational cross-sectional design, using validated questionnaires and objective assessments. The findings are clinically relevant, as they provide insights into the role of binocular vision dysfunction in digital eye strain, an increasingly common issue in modern life.
Overall, the manuscript is well-organized and methodologically sound, but some areas need further clarification and improvements in discussion and interpretation. Additionally, language and readability can be refined to enhance clarity.
- The title correctly describes the study’s focus, but it could be more concise. A suggestion:
“Validation of Binocular Vision Assessment Tools in Digital Eye Strain Syndrome: The DESIROUS Study”
The abstract is well-structured, summarizing the study’s background, methods, results, and conclusions.
Introduction
Provides strong background on DESS and binocular vision anomalies.
- Research gap is unclear—explicitly state what this study contributes to DESS research.
Methods
Study design and instrument validation are well-described.
- Missing details: No time frame for data collection and confounding factors (e.g., lighting, posture). Specify these aspects.
Results
Clearly presented with appropriate statistical tests.
- Observer variability is noted, but reasons for differences should be explained. No subgroup analysis—consider screen time, gender, or diagnostic differences as factors.
Discussion
Findings are compared with previous literature, highlighting the clinical significance of binocular vision dysfunction in DESS.
- Speculative causality—clearly state that this study identifies an association, not causation. Clinical implications need more depth—how should eye care professionals use these results?Limitations should be more explicitly stated (e.g., small sample size).
Conclusion
The conclusion accurately summarizes the key findings.
- It does not provide clear next steps for research. It does not highlight clinical applications. Include a statement on how these findings should guide future studies. Mention how these assessments could be used in clinical settings.
The manuscript is scientifically sound, but there are minor language and grammatical issues.
- Some sentences are overly long and complex, making them harder to follow. Occasional grammatical errors and awkward phrasing. Inconsistent use of abbreviations (e.g., DESS vs. DES). Simplify overly long sentences for better readability. Proofread for grammatical issues (consider a professional language editing service). Ensure consistent terminology and abbreviation use.
Reviewer 2 Report
Comments and Suggestions for Authors
Suggestions for Improvement
Simplify the Introduction: Narrow the focus to the key issues right from the start, cutting out some of the less essential background details.
Clarify Research Hypotheses: Clearly stating the study’s hypotheses early on would help readers grasp why combining subjective and objective measures is so important.
Expand the Sample Size: Although our pilot provided valuable insights, including a larger and more varied group would strengthen the study’s conclusions and make the findings more generalizable.
Review Questionnaire Items: A closer look at the DEQ-5 could help identify and remove problematic items, thereby improving its reliability.
Standardize Objective Measurements: To reduce variability—particularly in the BUT test—consider more rigorous training for examiners or implementing video-assisted methods for consistency.
Enhance Data Visualization: Presenting key findings with tables or graphs could help readers better understand discrepancies like those seen in screen time estimates and reliability metrics.
Include Long-Term Follow-up: Adding a longitudinal component could provide insights into how DESS symptoms evolve over time.
Explore Additional Tools: Incorporating other assessment instruments, such as the Symptom Questionnaire for Visual Dysfunctions, might offer a more comprehensive view of the condition.
